# Gold(I) ion and the phosphine ligand are necessary for the anti-*Toxoplasma gondii* activity of auranofin

C. I. Ma,[1] J. A. Tirtorahardjo,[2] S. S. Schweizer,[3] J. Zhang,[3] Z. Fang,[3] L. Xing,[4] M. Xu,[4] D. A. Herman,[5] M. T. Kleinman,[5] B. S. McCullough,[6] A. M. Barrios,[6] R. M. Andrade[1,2]

**ABSTRACT**  Auranofin, an FDA-approved drug for rheumatoid arthritis, has emerged as a promising antiparasitic medication in recent years. The gold(I) ion in auranofin is postulated to be responsible for its antiparasitic activity. Notably, aurothiomalate and aurothioglucose also contain gold(I), and, like auranofin, they were previously used to treat rheumatoid arthritis. Whether they have antiparasitic activity remains to be elucidated. Herein, we demonstrated that auranofin and similar derivatives, but not aurothiomalate and aurothioglucose, inhibited the growth of *Toxoplasma gondii in vitro*. We found that auranofin affected the *T. gondii* biological cycle (lytic cycle) by inhibiting *T. gondii's* invasion and triggering its egress from the host cell. However, auranofin could not prevent parasite replication once *T. gondii* resided within the host. Auranofin treatment induced apoptosis in *T. gondii* parasites, as demonstrated by its reduced size and elevated phosphatidylserine externalization (PS). Notably, the gold from auranofin enters the cytoplasm of *T. gondii,* as demonstrated by scanning transmission electron microscopy-energy dispersive X-ray spectroscopy (STEM-EDS) and Inductively Coupled Plasma-Mass Spectrometry (ICP-MS).

**IMPORTANCE**  Toxoplasmosis, caused by *Toxoplasma gondii*, is a devastating disease affecting the brain and the eyes, frequently affecting immunocompromised individuals. Approximately 60 million people in the United States are already infected with *T. gondii*, representing a population at-risk of developing toxoplasmosis. Recent advances in treating cancer, autoimmune diseases, and organ transplants have contributed to this at-risk population's exponential growth. Paradoxically, treatments for toxoplasmosis have remained the same for more than 60 years, relying on medications well-known for their bone marrow toxicity and allergic reactions. Discovering new therapies is a priority, and repurposing FDA-approved drugs is an alternative approach to speed up drug discovery. Herein, we report the effect of auranofin, an FDA-approved drug, on the biological cycle of *T. gondii* and how both the phosphine ligand and the gold molecule determine the anti-parasitic activity of auranofin and other gold compounds. Our studies would contribute to the pipeline of candidate anti-*T. gondii* agents.

**KEYWORDS**  auranofin, *Toxoplasma*, repurposing, drug discovery, gold

Toxoplasmosis, a devastating neurological and ocular disease caused by *Toxoplasma gondii,* disproportionally affects the unborn child or the immunocompromised individual. In the United States, it is estimated that approximately 60 million people are infected with *T. gondii* (1) and are at risk of developing toxoplasmosis. The first line of therapy for toxoplasmosis has not changed for the past 60 years, although the at-risk population has continued to grow (2–4). These compounds have activity that is restricted to actively dividing parasites and have significant bone marrow toxicity, discovering of new therapies a priority. Due to the scarcity of new treatments and their relevance

Address correspondence to R. M. Andrade, rmandra1@uci.edu.

The authors declare no conflict of interest.

See the funding table on p. 15.

to human health, the CDC designated toxoplasmosis as one of the neglected parasitic diseases in the United States in 2014 (5).

Repurposing FDA-approved drugs accelerates drug discovery for neglected parasitic diseases such as toxoplasmosis (6, 7). Recently, auranofin has emerged as a promising broad-spectrum antiparasitic agent, demonstrating activity against *Plasmodium falciparum* (8), *Schistosoma mansoni* (9), *Leishmania spp.* (10, 11), *Trypanosoma cruzi* (12), *Trichomonas vaginalis* (13–15), *Giardia lamblia* (14, 16), *Entamoeba histolytica* (17), *T. gondii* (18), free-living amoebas (19–21), *Onchocerca* (22), and Microsporidium (23) among other parasites of public health importance.

Auranofin is a gold(I) [Au (I)] compound that was used to treat rheumatoid arthritis (24). It is postulated that Au (I), which has a high affinity for dithiol groups, inhibits the thioredoxin reductase of the parasite leading to the accumulation of reactive oxygen species (ROS) (15, 18, 25), and is most likely responsible for its antiparasitic activity. Interestingly, older gold salts including aurothiomalate and aurothioglucose, which were used to treat rheumatoid arthritis before auranofin (26), also contain Au(I). Whether aurothiomalate and aurothioglucose have activity against *T. gondii* remains unknown. Should these compounds demonstrate anti-*T. gondii* activity, they would contribute to the pipeline of new anti-*T. gondii* candidates with known safety profiles for human use. Herein, we explored whether aurothiomalate, aurothioglucose, and other auranofin derivatives have activity against *T. gondii*.

While we have previously demonstrated that auranofin has activity against *T. gondii* infection *in vitro*, and *in vivo* (18), we do not know its mechanism of activity. In this manuscript, we characterized the effect of auranofin during the lytic cycle of *T. gondii* as well as the type of death induced by auranofin on the parasite and how its molecule of gold accumulated within *T. gondii* during its intracellular and extracellular stages.

## MATERIALS AND METHODS

### Host cell and parasite cultures

Parasites and human foreskin fibroblasts (HFF) were grown and maintained in Dulbecco's modified Eagle's medium supplemented with 10% fetal bovine serum (Hyclone), penicillin, and streptomycin (50 µg/mL each) and 200 µM glutamine (complete medium or D10). *T. gondii* RH parasites (National Institutes of Health AIDS Reference and Reagent Repository, Bethesda, MD) and RH tachyzoites expressing cytoplasmic yellow fluorescent protein (YFP+ RH) (M.J. Gubbels, Boston College, Boston, Massachusetts) (25) were maintained by serial passage in HFF monolayers at 37°C in a humid 5% $CO_2$ atmosphere.

### Drugs and chemicals

Auranofin (Enzo Life Sciences), aurothioglucose, aurothiomalate, and triethyl phosphine (Sigma Aldrich) were dissolved in 100% DMSO as a stock solution (5 mg/mL) and then diluted in D10 for final concentrations of 0.156–40 µM (in twofold increments). Derivatives of auranofin were provided by the Barrios laboratory (University of Utah) and synthesized as previously described (24). Stock solutions were prepared in DMSO and diluted in D10 for final concentrations ranging from 0.156–320 µM (in twofold increments).

### Plaque assay and IC$_{50}$ determination

We measured the inhibition of parasite growth by plaque assay as modified from a previously established method (27, 28). Confluent monolayers of HFF in six-well plates were inoculated with $1 \times 10^2$ parasites of RH *T. gondii*. Immediately after, each well received D10 containing specified concentrations of each compound. Plates were incubated undisturbed for 5 days at 37°C in a humidified incubator with 5% $CO_2$. On the fifth day, each well was rinsed once with phosphate-buffered saline (PBS), fixed with 4% paraformaldehyde (15 min at room temperature), and stained with crystal

violet for 2–3 min. Plaques (visualized as irregular clear areas of cell lysis against the violet background of intact host cells) were enumerated for each well. To generate the inhibition curve, plaque numbers (represented as a percentage of plaques relative to the no drug control) were plotted vs drug concentration. The $IC_{50}$ value was calculated from the inhibitor vs normalized response curve with variable slope. The results represented the average of three independent $IC_{50}$ assays ± standard error of the mean.

## Host cell viability and $TD_{50}$ determination

Confluent monolayers of HFF host cells (approximately $1 \times 10^4$) plated in clear bottom 96-well plates were treated with twofold dilutions of compounds of interest. After 5 days of incubation, 15 µL of dye solution (tetrazolium salt) was added to each well. The plates were incubated at 37°C for 4 hours; then, the reaction was stopped to solubilize the formazan product (Cell Titer 96 Nonradioactive Cell Proliferation Assay Promega). Absorbance at 570 Ⱡ ($A_{570}$) was recorded using a multimode microplate plate reader (Molecular Devices Spectramax i5, SoftMax Pro 7.0.2 Software). The $A_{570}$ absorbance reading is directly proportional to the number of viable cells. The values are presented as percentages of cell viability relative to the untreated controls (defined as 100% survival) and plotted against the log of compound concentrations. The log (auranofin derivative) vs cell viability curve with variable slope was generated to calculate $TD_{50}$ values.

## Invasion assays

We performed invasion assays with or without pre-treatment of *T. gondii* tachyzoites with auranofin, as modified from previously described assays (28–31). We used RH parasites expressing YFP (32). First, we determine the best concentration of auranofin to achieve stable host cell morphology and viability (HFF) while maximizing the effect of auranofin in *T. gondii* by titration of auranofin (doses ranging from 1 to 5 µM). The best auranofin dose fitting our criteria was 3 µM. Therefore, we performed a modified invasion assay using $2.5 \times 10^6$ parasites freshly lysed YFP+ RH parasites (29) that were filtered and centrifuged before each assay as follows:

a. Invasion assay without pre-treatment of *T. gondii* with auranofin: Confluent monolayers of HFF in an 8-chamber slide were inoculated with YFP+ RH parasites in D10 with DMSO (control) or auranofin at 3 µM in D10 for 30 min at 37°C and 5% CO2. The chamber slide was kept warm throughout the invasion assay by immersing it in a container with 37°C water.

b. Invasion assay with pre-treatment of *T. gondii* with auranofin: Similarly, freshly filtered YFP+ RH parasites were centrifuged and resuspended in either D10 with DMSO (control) or auranofin at 3 µM in D10 for 30 min before the invasion assay. Upon 30 min pre-treatment with auranofin, a confluent monolayer of HFF was inoculated with these pre-treated parasites for 30 min as described above.

After the 30-min invasion, all host cells were washed with PBS and fixed with 4% paraformaldehyde, blocked with 3% bovine serum albumin (BSA), and then incubated with anti-SAG-1 antibody (DG52) (David Sibley, Washington University in St. Louis) (4) for 1 hour, followed by incubation with goat anti-mouse IgG conjugated to Alexa-594 fluorophore (A19069, Thermo Fisher). For quantification, eight randomly selected fields (with a minimum of 300 parasites) were counted for each sample. Parasites that did not invade were stained red with mouse anti-SAG1 (extracellular YFP+/SAG1+). Intracellular parasites, YFP+/SAG1−, remained green. The invasion rate was calculated by dividing the number of intracellular parasites by the total number of parasites counted.

A paired *t*-test was used to determine the statistical difference between the control or auranofin treatment from three independent experiments.

## Replication assay

Confluent HFF monolayers were infected with $5 \times 10^5$ YFP+ RH parasites in coverslips. Following 1-hour incubation at 37°C to allow for invasion, the monolayer was rinsed with PBS, and D10 in DMSO (control) or auranofin containing (2.5 µM) media was added back. Coverslips were then incubated for 24 hours to allow for parasite replication. Coverslips were then fixed in 4% paraformaldehyde and mounted with DAPI-containing mounting media. Parasites were identified as DAPI+/YFP+ under fluorescence microscopy. The number of parasites per parasitophorous vacuole was counted across 10 random fields of view per coverslip. At least 100 parasitophorous vacuoles (PV) per condition were counted from three independent experiments. Results were grouped into the following categories according to the number of parasites per PV: 1, 2, 4, and >4 parasites. A chi-square trend test was used to determine the statistical difference between the PV category of the control and auranofin treatment from three independent experiments.

## Egress assay

The egress assay was modified from previously published methods to measure lactate dehydrogenase (LDH) content as a function of egress (host membrane damage induced by parasite egress) (33, 34). Confluent monolayers of HFF in a 24-well plate were inoculated with $5 \times 10^5$ parasites (freshly mechanically released by needle and filtered through a 3 µm filter) for 30 hours before each drug treatment in three technical triplicates. These infected HFF monolayers were rinsed with 37°C Ringer's solution (Ward's Science) three times and supplied with 5 µM calcium ionophore A23187 (ACROS), 57 µM Zaprinast (Alfa Aesar), 5 µM auranofin (Enzo Life Sciences), or DMSO control (all in 5 µl) for 40 min in 5% CO2 and 37°C. The culture supernatant was collected from wells and centrifuged at 4°C and 500 $g$ for 5 min. The solution was kept at 4°C and assayed immediately for the LDH content. The $A_{490}$ reading is directly proportional to a known amount of LDH (standard curve) (CytoTox 96 Nonradioactive Cytotoxicity Assay, Promega). Egress induced by each drug is presented as a relative value after subtracting control levels of LDH induced by DMSO alone (the solvent for all drugs included in this assay) from the levels of LDH induced by the drugs. A $t$-test to determine the statistical difference between the relative egress triggered by auranofin and A23187 or auranofin and Zaprinast from three independent experiments was used.

## Flow cytometry

A confluent monolayer of HFF was infected with purified RH parasites (freshly mechanically released by needle and filtered through a 3 µm filter). The monolayer remained undisturbed for 24 hours to allow for invasion and replication. On the day of treatment, extracellular parasites were mechanically released and harvested after differential centrifugation (35) at $35 \times g$ for 5 minutes to prevent parasite loss before processing. The supernatant was then collected and centrifuged at $1,350 \times g$ for 10 minutes to pellet the extracellular parasites. Parasites were plated in a 12-well plate ($2.7 \times 10^7$ per well) into 1 mL of D10 in DMSO (control) or auranofin (2.5 µM) containing media and incubated for 24–48 hours at 37°C with 5% CO2 before collection.

Parasite size after treatment with auranofin was determined by flow cytometry (forward scattering). At the end of each incubation time, parasites ($1 \times 10^5$) were resuspended in 1 mL of PBS to prepare them for flow cytometry.

Given the smaller size of parasites after auranofin treatment, we evaluated whether *T. gondii* death occurred by apoptosis *via* Annexin V assays (Invitrogen). After 24 hours of treatment, parasites were collected, washed with PBS, and resuspended in Annexin binding buffer (10 mM HEPES, 140 mM NaCl, and 2.5 mM CaCl2, pH 7.4) at a concentration of $10^6$ parasites per mL. One hundred microliters of this parasite suspension was stained with Annexin V-Alexa fluor conjugate before flow cytometry analysis. A $t$-test was used to determine the statistical difference between control or auranofin treatment groups from three independent experiments.

## Qualitative detection of gold by scanning transmission electron microscopy-energy dispersive X-ray spectroscopy

To study the uptake of gold (I) from auranofin by *T. gondii* tachyzoites, we performed scanning transmission electron microscopy-energy dispersive X-ray spectroscopy (STEM-EDS) of *T. gondii* parasites.

For intracellular parasites, we infected HFF monolayers with *T. gondii* overnight. Then, we incubated these monolayers with auranofin (10 µM) for 4 hours before purification of parasites (mechanical release, needling, and filtering to remove all host cell debris).

For extracellular parasites, we purified extracellular *T. gondii* tachyzoites (freshly mechanically released, needled, and filtered to remove host cell debris, $1 \times 10^7$) and treated them with auranofin (10 µM) for 4 hours incubated in suspension at 37°C with 5% $CO_2$ before collection.

After incubation, all parasites were centrifuged at $400 \times g$ for 10 min and washed with PBS twice. Parasite pellets were resuspended in 4% paraformaldehyde for 30 min and washed with PBS thrice. Lastly, the parasite pellets were resuspended in a final volume of 200 µL of PBS. A drop of fixed parasites was mounted in a Quantifoil grid. The parasites were either negatively stained with 2% uranyl acetate to confirm their morphology, while the parasites for EDS analysis were air-dried without staining with uranyl acetate to prevent overlapping of gold and uranium electron energy spectrums. These parasites were then analyzed under a JEM-2800 TEM/STEM microscope operated at 200 kV with a field emission electron source. The images were acquired at high-angle annular dark field (HAADF) imaging mode, and EDS spectra were collected with the large-angle dual dry solid state 100 mm$^2$ detectors. The gold signal was identified based on its characteristic emission at 9.7 keV.

## Quantitative detection of gold by inductively coupled plasma mass spectrometry

For intracellular parasites, *T. gondii* was allowed to infect confluent monolayers of HFF for 24 hours until >50% monolayer showed infected cells with parasitophorous vacuoles containing ≥8 parasites. This monolayer was exposed to auranofin (10 µM) for 4 hours before mechanically releasing the parasites, needling, and filtering before processing for gold quantification.

For extracellular parasites, *T. gondii* tachyzoites were purified (mechanically released, needled, and filtered to remove host cell debris) before treatment with auranofin (10 µM).

*T. gondii* tachyzoites ($1 \times 10^6$) were fixed with 4% paraformaldehyde and washed thrice with PBS before further processing for the quantification of gold.

The parasite samples were further washed with 1.5 mL of Nano-Pure water, followed by resuspension in 1.5 mL Aqua regia. Samples were transferred to 25 mL scintillation vials, heated to 200°C using a heat block, and digested for 2 hours. Glass marbles inside burette funnels were placed over the mouth of the scintillation vials allowing accumulated pressure to escape from the digestion vessel. Following the digestion period, we reduced the volume of samples to 0.1 mL and added hydrogen peroxide. The fully digested samples were diluted to a volume of 1.5 mL using Nano-Pure water and passed through a 0.1 µm pore filter. The samples were then further diluted to a final volume of 3 mL using Nano-Pure water. Inductively Coupled Plasma Mass Spectrometer (ICP-MS; Nu Instruments Attom, UK) was used to measure Gold (Au197); Rubidium (Rb85) was used as the internal standard. The ICP methods were optimized and validated before the analysis of the samples. The applied analytical methods enabled the determination of each selected element with satisfactory accuracy in addition to good precision (expressed by the coefficient of variation in terms of repeatability was in the range from 0.6% to 4.0 % for Au) and low limits of detection and quantitation (0.08 µg L$^{-1}$ and 0.09 µg L$^{-1}$ for Au, respectively). The measured concentrations of gold in the reference material were in good agreement with the certified values. Argon, with a purity of 99.999%, was used for plasma generation, nebulization, and as an auxiliary gas. All reagents used in this study

were of analytical reagent grade or the highest purity available. Ultrapure deionized water (18.2 MΩ), obtained from the WG-HLP deionizing system (Barnstead NanoPure), was used throughout. Single-element Au and Rb standards (LGC Standards USA) were used as calibration and control of all the analytical processes.

## Statistics and reproducibility

All biological replicates were performed with independently derived parasite populations. Continuous variable data were presented as the mean ± SD. Where indicated, statistical tests were performed on raw data, prior to normalization. Unpaired, two-tailed Student's *t*-test assessed the significance of differences. For categorical variables, a chi-square test for trend was calculated to determine if there was an association between categorical variables. Variable slope curves for $IC_{50}$ and $TD_{50}$ calculations were generated with GraphPad software. *P*-values of <0.05 were defined as significant. All statistical analyses were performed and visualized by GraphPad Prism 8 (GraphPad Software Inc., La Jolla, CA, USA). Tables and results were visualized in Excel (Microsoft). All representative experiments were performed at least thrice with comparable results.

## RESULTS

### Only auranofin and its derivatives inhibited *T. gondii* growth

Aurothiomalate, aurothioglucose, and auranofin all contain gold in the +1 oxidation state [Au(I)]. We hypothesized that they might have similar anti-*T. gondii* activity. To our surprise, neither aurothioglucose nor aurothiomalate inhibited the growth of *T. gondii in vitro* at any given concentration (0–40 µM) (Fig. 1A). These gold compounds differ in their ligand to stabilize Au(I) [auranofin contains a phosphine ligand (triethylphosphine) while aurothiomalate and aurothioglucose do not]. We therefore hypothesized that the phosphine ligand on auranofin (triethylphosphine) could be responsible for auranofin's inhibitory effect on *T. gondii* growth. However, triethylphosphine alone did not inhibit *T. gondii* growth (Fig. 1A). If the combination of Au(I) and phosphine ligand is required for the anti-*T gondii* activity of auranofin, we hypothesized that other auranofin derivatives generated with similar phosphine ligands, should also display anti-*T. gondii* activity. Selected auranofin derivatives 6, 10, 31, 39 with the general formula Cl–Au–$PR_3$ inhibited parasite growth with $IC_{50}$ values of 0.46 µM, 0.67 µM, 1.41 µM, and 0.59 µM, respectively (Fig. 7) (Fig. 1B).

### Auranofin derivative 31 had minimal effect on the viability of host cells

After observing that auranofin derivatives 6, 10, 31, and 39 had similar inhibitory activity to that of auranofin against *T. gondii*, we decided to evaluate whether they have similar cytotoxicity in host cells ($TD_{50}$). While auranofin had a $TD_{50}$ of 5.14 µM, auranofin derivatives 6, 10, and 39 had $TD_{50}$ of 4.87 µM, 4.53 µM, and 9.45 µM, respectively. Their corresponding *in vitro* therapeutic indexes, defined as the ratio of $TD_{50}$ on host cells over the $IC_{50}$ on parasites, were 10.6, 6.8, and 16, respectively. Remarkably, auranofin derivative 31 had a $TD_{50}$ 178 µM and an estimated *in vitro* therapeutic index of 126.2 (Fig. 1C and 7).

### Auranofin inhibited the invasion into host cells but not the replication of *T. gondii*

To understand the mechanism of activity of auranofin on *T. gondii*, we evaluated the impact of auranofin in each of the stages of *T. gondii*'s life cycle (lytic cycle): invasion, replication, and egress.

For the invasion stage, we performed modified invasion assays to quantify host cell invasion by subtracting extracellular parasites (parasites that did not invade or remained extracellular were stained in red) from the total number of parasites (green fluorescence). Without pre-treatment, the percentages of *T. gondii* that successfully invaded host cells

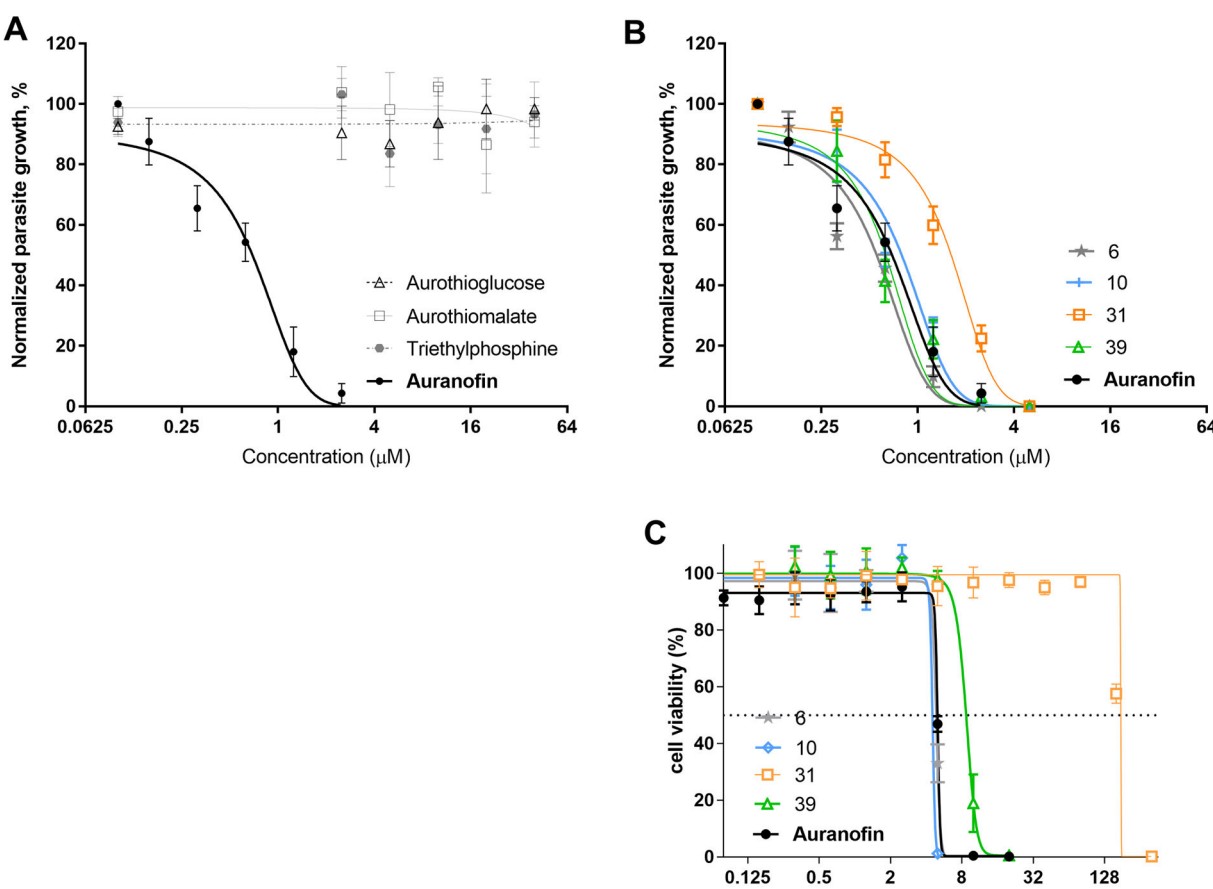

**FIG 1** A and B: $IC_{50}$ for gold compounds: HFFs were infected with $1 \times 10^2$ RH *T. gondii* parasites and incubated with different pharmacological agents for 5 days. The plaque formation in different drug concentrations was plotted as a growth curve, normalized against a no-drug control showing no inhibition. Growth curves are shown with mean inhibition concentration ± standard error. The $IC_{50}$ value was calculated from the inhibitor vs normalized response curve with variable slope. (A) $IC_{50}$ of aurothiomalate, aurothioglucose, and triethylphosphine against *T. gondii*: Aurothioglucose, aurothiomalate, and triethylphosphine did not display inhibitory activity against *T. gondii* up to a concentration of 40 µM. (B) $IC_{50}$ of auranofin derivatives against *T. gondii*: Compounds 6, 10, 31, and 39 inhibited *T. gondii* growth at similar $IC_{50}$ values (Fig. 7). (C) $TD_{50}$ of auranofin and auranofin derivatives: Host cell viability (HFF) was determined after 5 days of incubation with auranofin and auranofin derivatives 6, 10, 31, and 39. The percentage of cell viability is normalized to untreated controls and presented as mean viability ± standard error. Cell viability was plotted against the log of compound concentrations. Cell viability curves are shown with variable slope generated to calculate $TD_{50}$ values.

are similar in the control and auranofin treatment groups (59% vs 60%, respectively, $P = 0.7127$) (Fig. 2A). As seen in Fig. 2B, pre-treatment of extracellular *T. gondii* tachyzoites with auranofin decreased the invasion in the auranofin group compared to the control (18% vs. 35%; respectively, $P < 0.0154$).

For the replication stage, we infected host cells with *T. gondii* before treatment with auranofin. Once *T. gondii* invaded the host cell, auranofin treatment did not inhibit parasite replication, as we observed a similar number of parasites per parasitophorous vacuole (PV) in control and treated monolayers after 24 hours (Fig. 3A). This effect was not associated with abnormalities of the host cells since examination of the monolayer did not show changes in the monolayer integrity following 24 hours of auranofin treatment.

## Auranofin induced the egress of *T. gondii* from the host cell

The last stage of *T. gondii's* lytic cycle corresponds to its exit from the host cell or egress. Standard egress assays measure the content of LDH as an indicator of host membrane damage through parasite egress (34). Previously established egress-inducing agents such

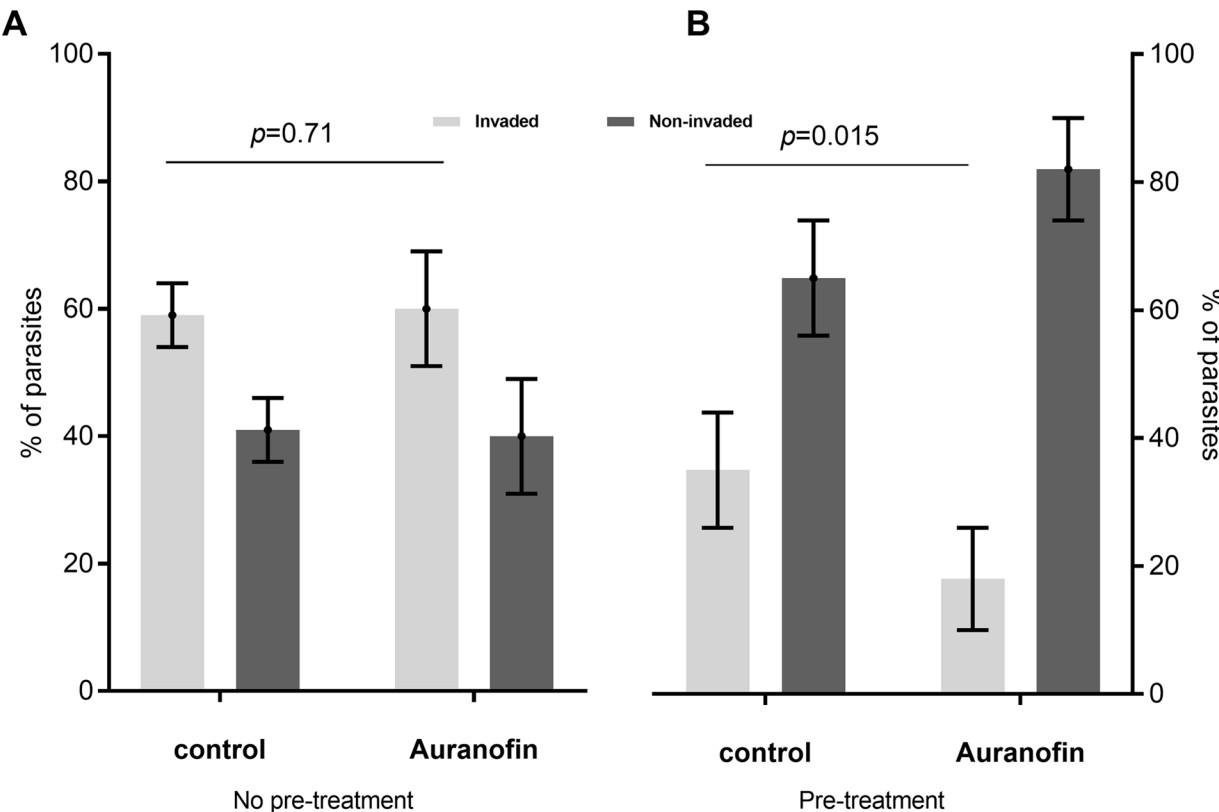

**FIG 2** Evaluation of the effect of auranofin during invasion: (A) Invasion rates of HFF in control and treatment groups without pre-treatment of *T. gondii* with auranofin before invasion: 59% vs 60%, respectively; *P* = 0.7127. (B) Invasion rates of HFF with parasites pre-treated with auranofin before invasion assay in control and treatment groups: 35% vs 18%, respectively; *P* < 0.0154. Results from three independent experiments. Error bars for standard deviations are shown.

as zaprinast, a phosphodiesterase inhibitor that induces egress by activating the parasite protein kinase G (PKG)(36), and A23187, a calcium ionophore that modifies intracellular calcium concentrations, triggers motility, and induces egress (37), were both included as a comparison.

After 30 hours of replication, subsequent treatment with each pharmacological agent for 40 minutes induced intracellular parasites to egress. The ionophore A23187 was a more potent egress inducer than auranofin (LDH concentrations 0.4533 ± 0.01148 vs 0.2768 ± 0.03435, respectively; *P* = 0.0082). Noteworthily, the amount of LDH release induced by auranofin is approximately 60% of that released by A23187. By contrast, auranofin and zaprinast released similar LDH concentrations (0.3272 ± 0.009 vs 0.2768 ± 0.034, respectively; *P* = 0.23212) (Fig. 3B).

## Auranofin induced the death of *T. gondii* by apoptosis

Our experiments showed that auranofin exerts its anti-*T. gondii* effect while the parasite is in the extracellular stages. As an obligate intracellular parasite, *T. gondii* is more susceptible to death in such a stage. Our experience working with auranofin and *T. gondii* suggests that pre-treatment of *T. gondii* with auranofin induces smaller parasite sizes. To validate our observation, we purified *T. gondii* parasites (mechanically released by needle and filtered through 3 µm membranes to remove host cell debris) before exposing them to auranofin for 24 hours. These parasites were processed for flow cytometry to measure their size by forward scatter (FSC-H). All parasites exposed to auranofin displayed smaller sizes than their corresponding controls (152.7 ± 8.225 *vs.* 201.2 ± 4.064, respectively; *P* = 0.0062). This effect continued as auranofin-treated parasites were smaller than their controls at 48 hours (123.5 ± 7.339 vs 169.9 ± 66.25, respectively, *P* = 0.0086) (Fig. 4A and B). These smaller parasites treated with auranofin showed increased phosphatidylserine

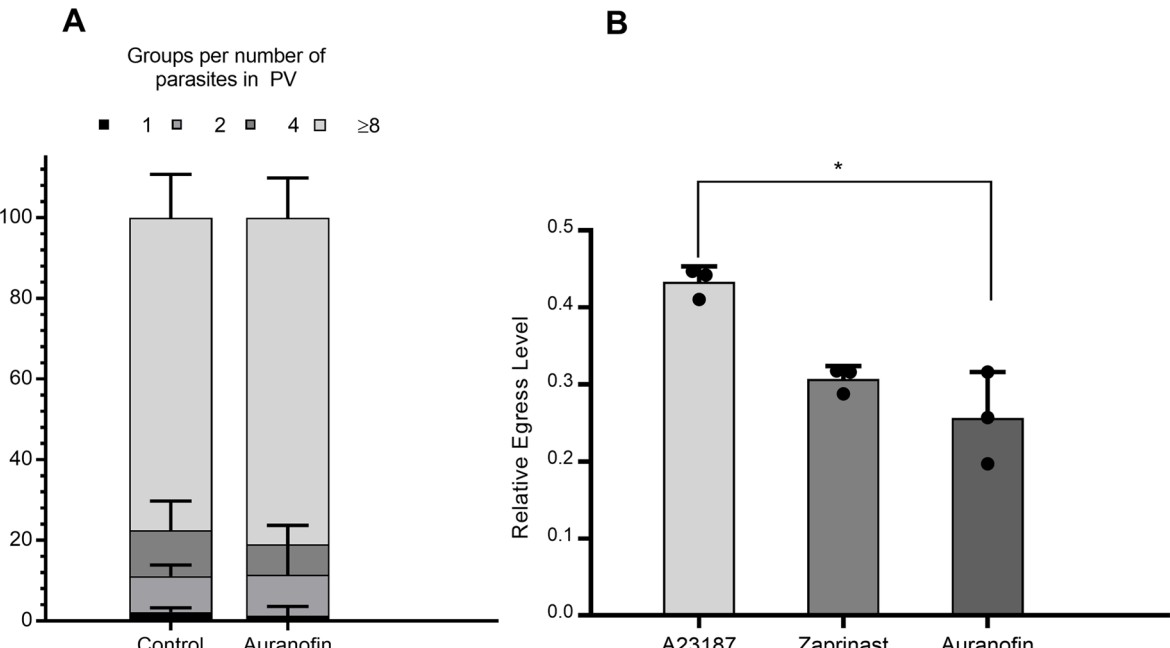

**FIG 3** Evaluation of the effect of auranofin during replication and egress of *T. gondii:* (A) Replication assays: Monolayers of HFF were infected with *T. gondii* parasites for 1 hour, then washed and D10 (control) or D10 containing auranofin (2.5 µM) (treatment) was added back for 24 hours. The number of parasites per parasitophorous vacuole (PV) was categorized according to the number of parasites per PV: 1, 2, 4, and ≥8 parasites. (*) 90-95% contained eight parasites per PV. A chi-square test for the trend between these categories (treatment and PV) was 2.435, $P = 0.4871$. Results are presented as percentages with SD from three independent experiments. (B) Egress assays: HFF were infected with *T. gondii* for 30 hours before exposure to A23187(5 µM), zaprinast (57 µM), auranofin (5 µM), and DMSO (solvent) for 40 minutes. Relative egress levels were normalized to DMSO (solvent): Auranofin 0.2768 ± 0.034, Zaprinast 0.3272 ± 0.009, and A23187 0.4533 ± 0.01148. The *t*-test between auranofin and Zaprinast egress levels showed no significant differences ($P = 0.23212$) while they were significantly lower than those of A23187 ($P = 0.0082$). Results from three independent experiments. Error bars for standard deviations are shown.

externalization (PS) detected by Annexin V assay compared to controls (29.16 ± 3.615 vs 11.2 ± 2.937; $P = 0.0182$) at 24 hours (Fig. 4C). Beyond 24 hours, low parasite recovery limited our attempts to perform Annexin V assays.

## Gold from auranofin entered *T. gondii* parasites

Auranofin's activity against *T. gondii* occurs during invasion and egress when the parasite is in the extracellular stage. Since gold has a high affinity for thiol groups, abundantly present in eukaryotic cells such as the host and the parasites, we hypothesized that gold distribution in the parasite is higher during extracellular stages. To test our hypothesis, we worked with parasites treated with auranofin while replicating inside the host cells (intracellular) and parasites treated with auranofin during the extracellular stage. These parasites were purified (needled and filtered through a 3 µm membrane) to remove host cell debris before processing them for STEM-EDS. STEM-EDS detected gold as an emission peak at 9.7 keV within intracellular and extracellular parasites (Fig. 5A right lower panels), demonstrating that auranofin's gold can enter the parasite during both stages. This peak appears larger during extracellular stages (Fig. 5A right panel, bottom). These observations were quantified by ICP-MS, showing higher concentrations of gold in extracellular parasites (7.181ppb ±0.4251, $n = 6$) compared to intracellular parasites (3.701ppb ±0.2509, $n = 6$) ($P < 0.0001$) (Fig. 5B).

## DISCUSSION

We have previously demonstrated that auranofin has activity against *T. gondii in vitro* and *in vivo* (18). However, the mechanism by which auranofin exerted anti-*T. gondii* activity and the possibility that other gold compounds might have similar activity had not been

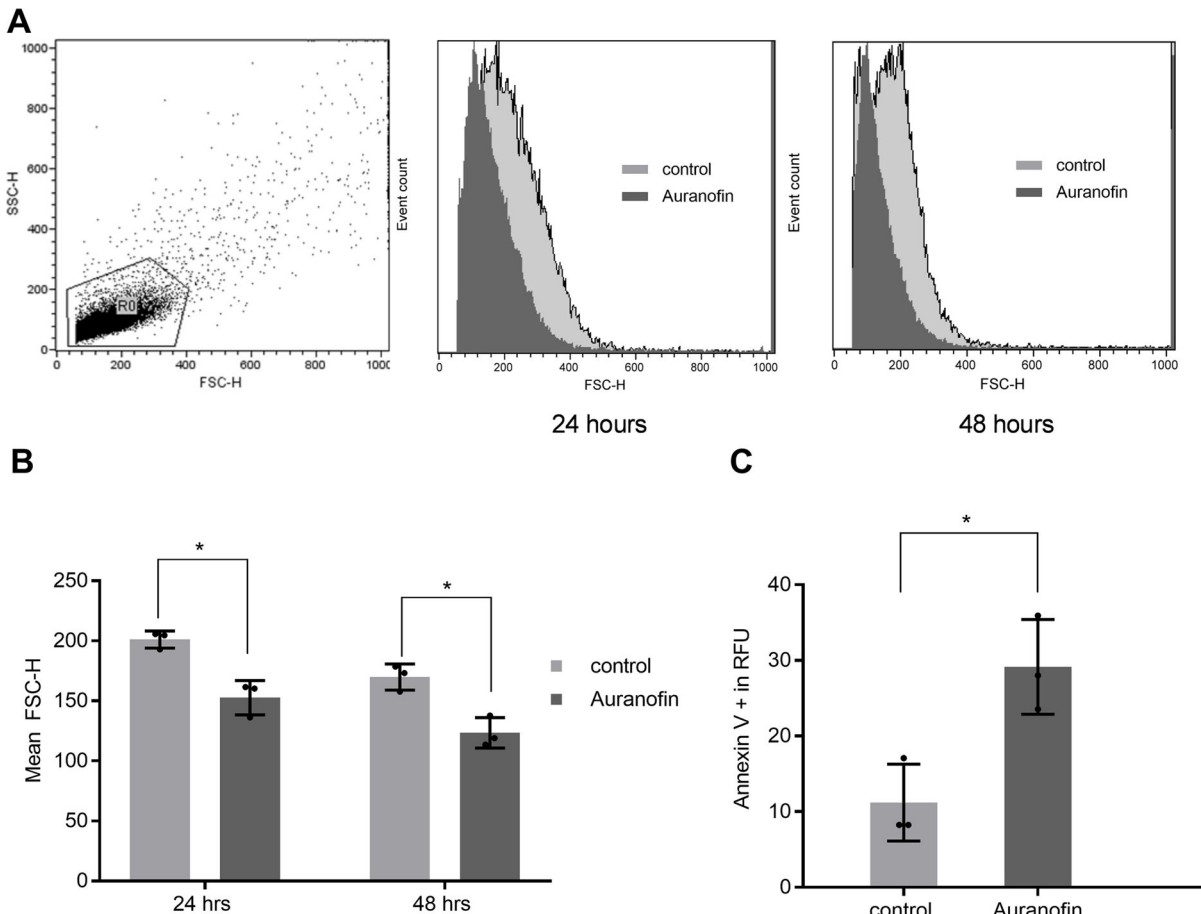

**FIG 4** Apoptosis of *T. gondii* by auranofin: Purified *T. gondii* parasites were exposed to auranofin for 24 hours or 48 hours before preparing them for flow cytometry. (A) Size of *T. gondii* parasites as determined by FSC-H: Gating strategy of parasites (left); FSC-H for control (light gray); and auranofin-treated parasites (dark gray) at 24 hours (center) and 48 hours (right). (B) Evaluation of *T. gondii* parasite size: Auranofin-treated parasites were smaller than those in the control group at 24 hours (152.7 ± 8.225 vs 201.2 ± 4.064, respectively; $P = 0.0062$) and at 48 hours (123.5 ± 7.339 vs 169.9 ± 66.25 respectively; $P = 0.0086$). (C) Phosphatidylserine externalization (PS) as demonstrated by Annexin V assay and presented as relative fluorescent units: auranofin-treated 29.16 ± 3.615; and control parasites: 11.2 ± 2.937; $P = 0.0182$. Results from three independent experiments. Error bars for standard deviations are shown.

investigated. Here, we showed that auranofin inhibits *T. gondii* invasion of the host cell and triggers *T. gondii* egress from the host cell but does not block replication of the parasite within a host cell. Exposure of *T. gondii* to auranofin triggers apoptosis in the parasite, which could be explained by the higher accumulation of gold during the extracellular stage of the parasite. Interestingly, the injectable gold salts aurothiomalate and aurothioglucose do not have anti-*T gondii* activity, while auranofin analogs did show good activity. These data, combined with the observation that triethylphosphine alone does not have any anti-*T gondii* activity, indicate that an Au(I) phosphine complex is required for activity.

Auranofin, an FDA-approved drug for treating rheumatoid arthritis, has shown promising broad-spectrum antiparasitic activity (8–16, 19–23, 38) including anti-*T. gondii* activity, as we have previously demonstrated (18, 25). Its chemical structure includes a gold(I) ion, a 2,3,4,6-tetra-O-acetyl-1-thio-β-D-glucopyranose ligand, and a triethylphosphine ligand (Fig. 6 and 7). It is considered a pro-drug (39) that delivers gold to dithiol groups, such as those in proteins bearing thioredoxin-containing domains (40). Hypothetically, gold(I) is responsible for the antiparasitic effects of auranofin. Herein, we investigated other gold salts that have been previously used to treat rheumatoid arthritis for activity against *T. gondii*. Like auranofin, aurothioglucose and aurothiomalate contain gold(I) (26). However, they are seldom recognized for their antimicrobial activity (41, 42).

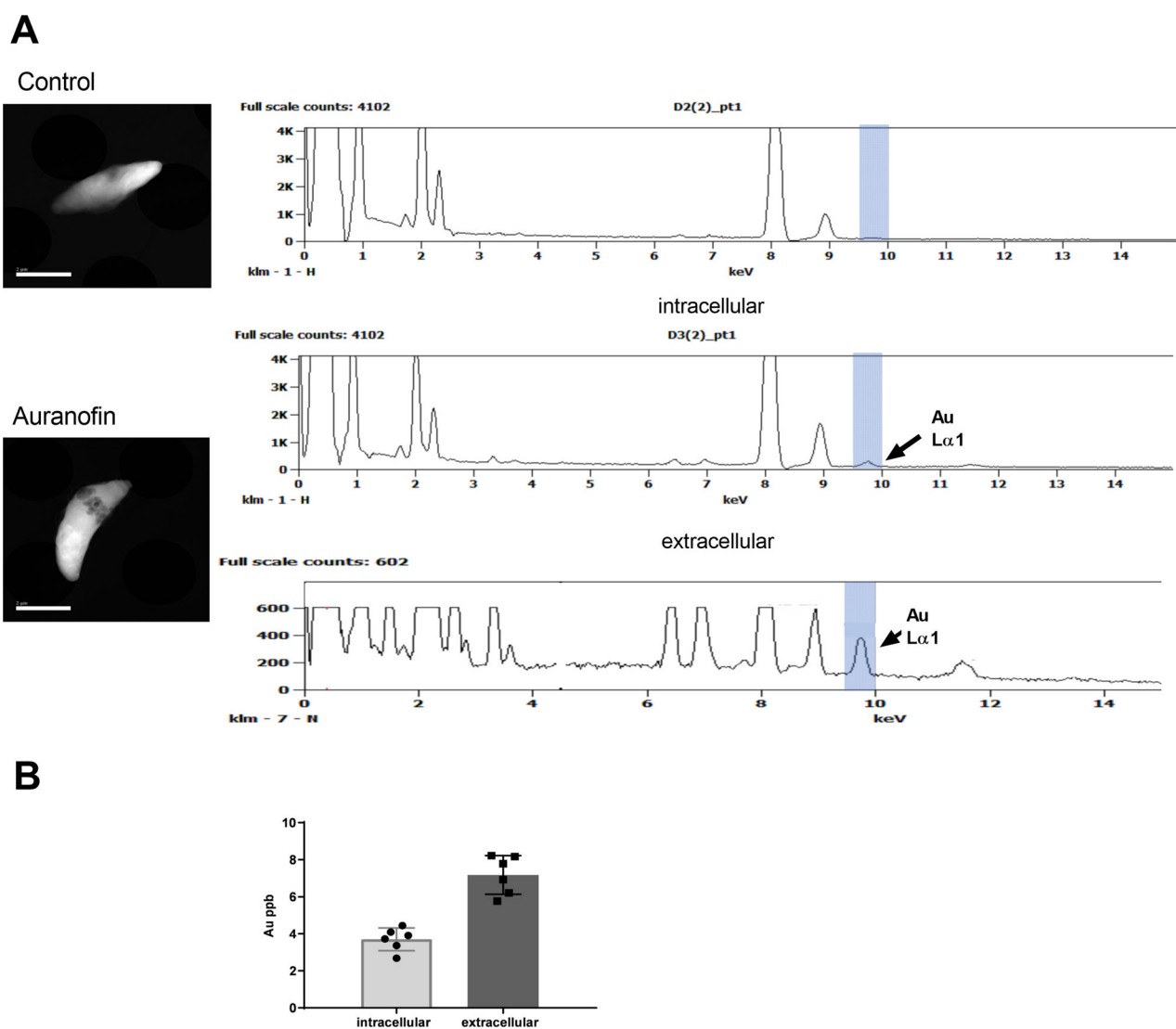

**FIG 5** Detection of gold in *T. gondii* tachyzoites: Purified *T. gondii* parasites (needled and filtered with 3 μm membrane to remove all host cell debris), were treated with 10 μM auranofin for 4 hours before processing. (A) Detection by energy Dispersion Spectrometry (EDS): Typical morphology of *T. gondii* tachyzoites, crescent shape and approximately 5 μm × 2 μm, was detected on high-angle annular darkfield (HAADF) images after uranyl acetate staining (*left* panels, top and bottom). Bar = 2 μm. The gold signal was identified by its characteristic emission at 9.7keV (AuLα1) *(right panels, bottom)*. (B) Detection by ICP-MS. Extracellular *T. gondii* parasites had a gold concentration of 7.181ppb ±0.4251, while intracellular parasites (exposed to auranofin while replicating inside the host cells) had a gold concentration of 3.701ppb ±0.2509 (*P* < 0.0001). Results from three independent experiments. ppb = parts per billion.

This lack of reports contrasts with the abundance of evidence supporting the broad antimicrobial activity of auranofin against a vast array of bacteria, fungi, and parasites. Our experiments showed that neither aurothioglucose nor aurothiomalate, despite their containing gold(I), had anti-*T. gondii* activity (Fig. 1). These gold salts differ from auranofin due to their lack of a phosphine ligand (Fig. 6). A phosphine ligand helps stabilize the oxidation state of the Au(I) ion, contributes to the lipophilicity of the compound (43), and potentially contributes to the antiparasitic activity of auranofin since aurothiomalate and aurothioglucose both lack such ligand and lack anti-*T. gondii* activity. It is reasonable to consider whether triethylphosphine, the phosphine ligand in auranofin, would inhibit *T. gondii* growth independently. However, our experiments showed that triethylphosphine did not exhibit anti-*T. gondii* activity (Fig. 1A). Since neither Au(I) nor triethylphosphine is sufficient for anti-*T. gondii* activity, we next investigated the activity of other Au(I)-phosphine complexes. Selected auranofin

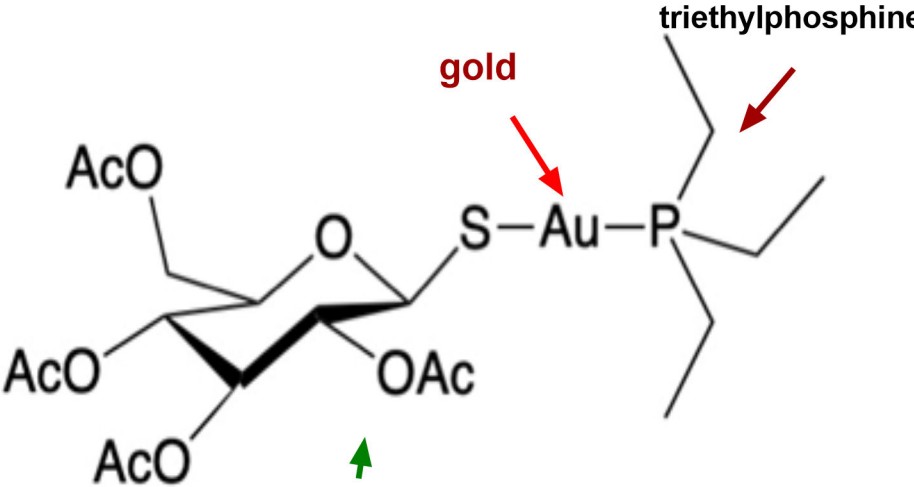

**FIG 6** Structure of auranofin.

derivatives (24) 6, 10, 31, and 39 showed activity *in vitro* against *T. gondii* with similar $IC_{50}$ to that of auranofin (Fig. 7). Most notably, derivative 31 displayed the highest *in vitro* therapeutic index, the ratio between $TD_{50}$ on host cells and $IC_{50}$ on the parasites. This feature is highly desirable for drug development to minimize host cytotoxicity while maintaining antiparasitic activity.

Compounds 39, 31, 6, and 10 are all gold(I) chloride complexes with the structure $Cl–Au–PR_3$. Compounds 6 and 31 are quite similar to one another; both have three 6-membered aromatic rings attached to the phosphorus in the ligand. In 31, all three functional groups (R groups) on the phosphorus are phenyl (an aromatic 6-membered ring, C6H5), while in 6, there are two phenyl rings and one benzyl ring (C6H5-CH2) (Fig. 7). Interestingly, compound 6 demonstrated antiparasitic activity against extracellular and intracellular forms of *Leishmania spp.* (promastigotes and amastigotes, respectively) at a similar $IC_{50}$ and $TD_{50}$ to that of auranofin, as shown by Sharlow et al. (11). Compound 31 also had activity against *Leishmania spp.* although at higher $IC_{50}$ than auranofin and was less toxic to host cells than auranofin and compounds 6, 10, and 39 (11). These observations suggest that the phosphine ligand in the $Cl–Au–PR_3$ complex is crucial for the interaction with biological targets in the parasite and the host, governing their ultimate antiparasitic activity and toxicity in the host cell.

How auranofin affects the life cycle (lytic cycle) of *T. gondii* remains poorly understood. Herein, we showed that auranofin inhibits *T. gondii's* ability to invade the host cells (Fig. 2B) only if *T. gondii* is exposed to auranofin before invasion. Auranofin also induces *T. gondii* egress from the host cell (Fig. 3B) but it cannot prevent replication once the parasite has entered the host cell (Fig. 3A). This effect of auranofin on *T. gondii's* life cycle could be explained by the postulated mechanism of activity of auranofin. Auranofin is known for its ability to induce the accumulation of reactive oxygen species (ROS) in eukaryote cells such as mammalian host cells (44) and *T. gondii* parasites (25). *T. gondii* is known for its ability to inhibit the ROS attack of the host cells to achieve successful intracellular invasion (45); however, the parasite is highly susceptible to ROS during extracellular stages as demonstrated by Camps and Boothroyd (46). Not surprisingly, auranofin's anti-*T. gondii* activity appears focused on the extracellular stages of the parasite when it is the most vulnerable to the effects of ROS accumulation. This vulnerability to oxidative damage by ROS is associated with the direct effect of gold in *T. gondii's* mitochondrion. As demonstrated by Adeyemi et al., gold nanoparticles have activity against *T. gondii* by inducing oxidative stress and reducing its mitochondrion membrane potential (47). Noteworthily our previous studies, through chemical

| Compound Number | Compound Structure | Molecular Weight | $TD_{50}$ | $IC_{50}$ | *In vitro* Therapeutic Index |
|---|---|---|---|---|---|
| 39 | Cl—Au—P | 350.58 | 9.45 | 0.59 | 16 |
| 31 | Cl—Au—P | 494.71 | 178 | 1.41 | 126.2 |
| 6 | Cl—Au—P | 508.74 | 4.87 | 0.45 | 10.06 |
| 10 | Cl—Au—P | 500.76 | 4.53 | 0.67 | 6.8 |
| Auranofin | AcO...S—Au—P | 679.5 | 5.14 | 0.56 | 9.2 |
| Aurothioglucose | HO...S—Au | 392.18 | N/A | N/A | N/A |
| Aurothiomalate | S—Au...Na⁺ | 390.07 | 141 | N/A | N/A |

**FIG 7** $TD_{50}$ and $IC_{50}$ of gold-containing compounds.

mutagenesis of *T. gondii* parasites treated with auranofin (25), found mutations in *T. gondii's* mitochondrial superoxide dismutase, suggesting that auranofin's target in *T. gondii* could be located in the parasite's mitochondrion. This parasite's organelle is highly susceptible to oxidative damage during nutrient starvation (48, 49) or treatment with drugs such as monensin (50). In addition, *T. gondii* extracellular stages depend more on

oxidative phosphorylation for ATP generation than its intracellular forms (51), therefore more dependent on its mitochondrion.

The ultimate death of the parasite by auranofin seems to occur by induction of apoptosis in *T. gondii*. Extracellular parasites treated with auranofin demonstrated smaller size and increased phosphatidylserine expression, known markers of apoptotic death in eukaryotic cells, as determined by an Annexin V assay (Fig. 4). Similar results have been observed by Sharlow et al. with auranofin and its derivatives in *Leishmania spp.* (11). This mechanism of death differs from other inducers of oxidative stress in *T. gondii* such as monensin, which induces death in the parasites by autophagy (50).

Since auranofin did not demonstrate any effect on the replication of the parasite once *T. gondii* entered the host (Fig. 3A), we hypothesized that the lack of effect from auranofin on *T. gondii* replication was due to a lack of gold accumulation in the parasite during its intracellular stages. To test our hypothesis, we measured gold qualitatively and quantitatively (EDS-STEM and ICP-MS). As we have shown, gold reaches the intracellular space *of T. gondii* whether the parasite is inside or outside the host cell. However, gold concentrations were higher when *T. gondii* was exposed to auranofin during extracellular stages (Fig. 5). This differential distribution of gold is likely due to Au(I) ions being sequestered by eukaryotic dithiol groups, resulting in less overall gold available to enter the parasite. It is likely that the amount of gold accumulated in the parasite during intracellular stages was not sufficient to inhibit its target in the parasite.

While this lower accumulation of gold inside *T. gondii* during intracellular stages seemed insufficient to prevent further parasite replication (Fig. 3A), it was sufficient to induce *T. gondii* egress from the host cell (Fig. 3B). LDH release caused by auranofin treatment was 60% that of A23187 treatment, a known inducer of *T. gondii* cell egress (Fig. 4B). The doses of auranofin that we worked with do not affect the host cell monolayer during the time of experimentation (*data not shown*); therefore, these LDH concentrations reflect the direct effect of auranofin on *T. gondii* egress. How auranofin induces egress of the parasite from the host cell remains to be elucidated.

Our study is focused on *in vitro* observations and does not explore the effect of auranofin in the latent forms of the parasite and whether the effect of auranofin derivatives, such as compound 31, can be extrapolated to *in vivo* models of toxoplasmosis. Studies to answer these questions are currently in progress.

In summary, we characterized the impact of auranofin on the cell cycle of *T. gondii* and demonstrated that the overall effect of auranofin is more pronounced during the extracellular stages of the parasite. Our results also demonstrated that the presence of Au(I) alone is not enough to provide anti-*T. gondii* activity without the addition of an appropriate phosphine ligand. Based on the compounds tested in this manuscript, we propose that the chemical structure of candidate, anti-*T. gondii* compounds, is critical to inhibit parasite growth, induce parasite egress from the cell, and protect the viability of the host cells. Further studies to determine whether promising auranofin derivatives can reach the central nervous system of the host without added cytotoxicity are part of our future directions. Repurposing drugs that have been proven safe for humans is an alternative approach to speed up the process of drug discovery for neglected parasitic diseases such as toxoplasmosis. Understanding how these repurposed drugs work and affect the parasite of interest is critical for further drug development.

## ACKNOWLEDGMENTS

Special gratitude to Dr. Melissa Lodoen and Dr. Naomi Morrissette (University of California Irvine) for their advice and critical review of this article. We also thank Dr. David Sibley (Washington University in St. Louis Missouri) for sharing the DG52 antibody for our invasion assays, and Dr. Marc Jean M.J. Gubbels (Boston College, Boston, Massachusetts) for sharing *T. gondii* YFP.

R.M.A. was supported by NIHK08 5K08AI102989-04, AMFDP RWJF 70642 (the views expressed here do not necessarily reflect the views of the Foundation), and UCI DOM Chair Research Award 60244.

The authors declare that the research was conducted without any commercial or financial relationships that could be construed as a potential conflict of interest.

## AUTHOR AFFILIATIONS

[1]Department of Medicine, Division of Infectious Diseases, University of California at Irvine, Irvine, California, USA

[2]Department of Microbiology and Molecular Genetics, University of California at Irvine, Irvine, California, USA

[3]School of Biological Sciences; University of California at Irvine, Irvine, California, USA

[4]Irvine Materials Research Institute; University of California at Irvine, Irvine, California, USA

[5]Department of Medicine, Occupational and Environmental Medicine, University of California at Irvine, Irvine, California, USA

[6]Department of Medicinal Chemistry, University of Utah College of Pharmacy, Salt Lake City, Utah, USA

## PRESENT ADDRESS

S. S. Schweizer, Benchling, San Francisco, California, USA

J. Zhang, Stanford University, Stanford, California, USA

Z. Fang, CSC Family Health, Los Angeles, California, USA

B. S. McCullough, Salk Institute for Biological Studies, La Jolla, California, USA

## AUTHOR ORCIDs

R. M. Andrade http://orcid.org/0000-0002-8802-1118

## FUNDING

| Funder | Grant(s) | Author(s) |
| --- | --- | --- |
| HHS \| National Institutes of Health (NIH) | NIHK08 5K08AI102989-04 | R. M. Andrade |
| Robert Wood Johnson Foundation (RWJF) | AMFDP RWJF 70642 | R. M. Andrade |

## AUTHOR CONTRIBUTIONS

C. I. Ma, Data curation, Formal analysis, Investigation, Methodology, Project administration, Supervision, Writing – original draft, Writing – review and editing | J. A. Tirtorahardjo, Data curation, Formal analysis, Investigation, Methodology, Supervision, Writing – original draft, Writing – review and editing | S. S. Schweizer, Data curation, Formal analysis, Investigation, Methodology, Validation | J. Zhang, Data curation, Formal analysis, Investigation, Methodology | Z. Fang, Data curation, Formal analysis, Investigation, Methodology, Project administration, Supervision, Writing – original draft, Writing – review and editing | L. Xing, Data curation, Formal analysis, Investigation, Methodology, Supervision, Validation, Visualization, Writing – original draft, Writing – review and editing | M. Xu, Data curation, Formal analysis, Investigation, Methodology, Validation, Visualization | D. A. Herman, Data curation, Formal analysis, Investigation, Methodology, Writing – original draft | M. T. Kleinman, Data curation, Formal analysis, Investigation, Methodology | B. S. McCullough, Data curation, Formal analysis, Investigation, Methodology, Resources, Validation, Visualization, Writing – original draft, Writing – review and editing | A. M. Barrios, Conceptualization, Data curation, Formal analysis, Investigation, Methodology, Resources, Validation, Visualization, Writing – original draft, Writing – review and editing | R. M. Andrade, Conceptualization, Data curation, Formal analysis, Funding acquisition, Investigation, Methodology, Project administration, Resources, Supervision, Validation, Visualization, Writing – original draft, Writing – review and editing

## ADDITIONAL FILES

The following material is available online.

### Open Peer Review

**PEER REVIEW HISTORY (review-history.pdf).** An accounting of the reviewer comments and feedback.

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
