## [Reviewer comments · Microbiology Spectrum]

Microbiology Spectrum

Gold(I) ion and the phosphine ligand are necessary for the anti-*Toxoplasma gondii* activity of Auranofin

Christopher Ma, James Tirtorahardjo, Sakura Schweizer, Jerry Zhang, Zian Fang, Li Xing, Mingjie Xu, David Herman, Michael Kleinman, Brandon Mccullough, Amy Barrios, and Rosa Andrade

Corresponding Author(s): Rosa Andrade, University of California Irvine School of Medicine

Review Timeline:

Submission Date:	August 1, 2023
Editorial Decision:	September 25, 2023
Revision Received:	December 7, 2023
Accepted:	December 8, 2023

Editor: Brice Rotureau

Reviewer(s): The reviewers have opted to remain anonymous.

Transaction Report:

DOI: <https://doi.org/10.1128/spectrum.02968-23>

September 25, 2023

Dr. Rosa M. Andrade
University of California Irvine School of Medicine
Medicine
1001 Health Sciences Rd
Medical Sciences Building 821C
Irvine, CA 92697

Re: Spectrum02968-23 (**Gold(I) ion and the phosphine ligand are necessary for the anti*Toxoplasma gondii* activity of Auranofin**)

Dear Dr. Rosa M. Andrade:

Your manuscript was evaluated by two independent reviewers. Most of their comments are dwelling upon technical aspects that should be considered and that may require additional experiments.

Link Not Available

Sincerely,

Brice Rotureau

Journals Department
Reviewer comments:

Reviewer #1 (Comments for the Author):

Auranofin is previously reported to have potent inhibitory effect on several parasites, including *Toxoplasma*. This study demonstrated the in vitro anti- *T. gondii* effect of auranofin and derivatives, and tried to dissect the mechanism of action of auranofin. The manuscript is well organized and written, the results are of some interest. However, there are several major concerns need to be addressed.

1. For the methodology of IC50 determination, this study relies on counting the plaque numbers to generate the inhibition curves. Actually, more accurate and also very common way is to detect the luminance of a reporter gene expressing parasite line, such as GFP or luciferase.
2. Considering the potential anti-*T. gondii* effect and the structural variation of the derivatives of auranofin, the derivatives are necessary to be included in the experiments that demonstrate their mode of action.
3. The study showed that auranofin treatment induced apoptosis in *T. gondii*, but how the apoptosis was happened or by which pathway is not determined. RNA-seq after treatment may be a good way to find some clues for this. And immunostaining or immunoblotting assays of specific factors may help for the validating.
4. Figure 3A: for the replication assays, >4 parasites per PV need to be specified, eg 8, 16.

Reviewer #2 (Comments for the Author):

Ma et al. reported the detailed analysis on the anti-Toxoplasma activity of auranofin. By testing similar gold ion containing compound, they showed that Au(I) and phosphine complex is required for the activity, and the extracellular parasites was affected, whereas the intracellular replication was not affected. Experimental designs are sound, and data and statistical analysis supported their conclusions. Followings need to be clarified to avoid the misleading.

Page 4, Line 73,

Trichomona vaginalis should be *Trichomonas vaginalis*.

Page 5, Line 88,

Citation number (21) may be (18). Please check and correct if required.

Page 6, Line 116-118,

Please clarify that parasite were inoculated first and the medium was changed after that or parasites were mixed with drug containing medium and was placed onto host cells. As they showed that invasion ratio decreases with pre-treatment of the extracellular parasites with auranofin, the time point of the initiation of drug treatment is important. The plaque number can be affected by the initial invasion, while later replication, egress, re-invasion also affects whether it becomes in visible size or not. The size of the plaque is highly recommended to be shown as an indicative value for the parasitic growth.

Page 7, Line 140

Please unify the terms to be used. Toxic dose (TD50) or Lethal concentration (LC50)[Table 1]

Page 10, Line 203,

Does normalization mean that A450 values of DMSO conditions were subtracted from the treated group A450 values? Please clarify what relative egress level represents.

Page 15, Line 318

While text and table write the auranofin derivative 6 had IC50 5uM, From Fig 1B derivative 6 did not have anti-parasitic effect upto 40 uM. Please clarify.

Page 16, Line 329

Do you mean that "Auranofin inhibited the invasion into host cells but not the replication of *T. gondii*"?

Page 16, Line 336

Methods sections wrote that the invasion rate was calculated as (invaded parasites / total parasites). Or you counted the number of the host cells which had invaded parasites per total host cells? Whichever value should reflect the successful invasion event, but please clarify which is the description of what you did.

Figure 1.

Some data points seems to be missing error bars. If the experiments for the aurothioglucose, aurothiomalate, Triethylphosphine in (A), and derivative 6 in (B) was performed in single replicate, please modify the text.

Figure 3A

If the parasites number per vacuole are shown in raw counting numbers rather than percentage in the total pv counts, please describe how the three independent experiments are shown in this figure. Is it the representative experiments, or just the sum up of the counts from each replicates?

Staff Comments:

Preparing Revision Guidelines

To submit your modified manuscript, log onto the eJP submission site at <https://spectrum.msubmit.net/cgi-bin/main.plex>. Go to Author Tasks and click the appropriate manuscript title to begin the revision process. The information that you entered when you first submitted the paper will be displayed. Please update the information as necessary. Here are a few examples of required

updates that authors must address:

Please return the manuscript within 60 days; if you cannot complete the modification within this time period, please contact me. If you do not wish to modify the manuscript and prefer to submit it to another journal, please notify me of your decision immediately so that the manuscript may be formally withdrawn from consideration by Microbiology Spectrum.

POINT by POINT RESPONSES TO REVIEWERS COMMENTS

Reviewer #1 (Comments for the Author):

Auranofin is previously reported to have potent inhibitory effect on several parasites, including *Toxoplasma*. This study demonstrated the in vitro anti- *T. gondii* effect of auranofin and derivatives, and tried to dissect the mechanism of action of auranofin. The manuscript is well organized and written, the results are of some interest. However, there are several major concerns need to be addressed.

1. For the methodology of IC50 determination, this study relies on counting the plaque numbers to generate the inhibition curves. Actually, more accurate and also very common way is to detects the luminance of a reporter gene expressing parasite line, such as GFP or luciferase.

We appreciate the reviewer's observation about using a fluorescent or luminescent parasite to quantify growth inhibition. It will be an assay to consider for future investigations since it will require additional time and costs for optimization.

Our plaque assays remain a valuable research tool widely used by the *Toxoplasma* research community due to their low cost, ease of optimization, and training, which we have standardized in my research group.

2. Considering the potential anti-*T. gondii* effect and the structural variation of the derivatives of auranofin, the derivatives are necessary to be included in the experiments that demonstrate their mode of action.

Certainly, each auranofin derivative deserves further investigation. However, it is beyond the scope of this manuscript. In fact, compound 31, which showed the greatest estimated therapeutic index, is part of our future investigations.

3. The study showed that auranofin treatment induced apoptosis in *T. gondii*, but how the apoptosis was happened or by which pathway is not determined. RNA-seq after treatment may be a good way to find some clues for this. And immunostaining or immunoblotting assays of specific factors may help for the validating.

We agree with the reviewer's suggestion on performing RNA-seq to characterize further the death phenotype of our auranofin-treated *T. gondii* parasites. As a first step to explore this recommendation, we attempted to perform qPCR to characterize the expression of several known markers of *T. gondii* death as previously described by Nyoman&Luder (PMID23468121). Our qPCR results did not find differences between *T. gondii* autophagy and apoptosis markers. These experiments require better-quality RNA, which we were not able to get due to the loss of

parasites after treatment with auranofin. While this experimental hurdle is disappointing, it is also reassuring to know that auranofin kills *T. gondii* consistently. We appreciate the reviewer's request as it became another of our future directions for the future projects.

4. Figure 3A: for the replication assays, >4 parasites per PV need to be specified, eg 8, 16.

Since the replication time of *T. gondii* is 7-8 hours, we did not expect to see any PV with 16 or more parasites in a replication assay that took 24 hours. In fact, 90-95% PV containing >4 parasites, had 8 parasites. PV with ≥ 16 parasites were extremely rare as it is expected per *T. gondii* biological cycle. As suggested by the reviewer, we clarified that observation in Figure 3A and legend with an asterisk (*): virtually all PV contained 8 parasites.

Reviewer #2 (Comments for the Author):

Ma et al. reported the detailed analysis on the anti-Toxoplasma activity of auranofin. By testing similar gold ion containing compound, they showed that Au(I) and phosphine complex is required for the activity, and the extracellular parasites was affected, whereas the intracellular replication was not affected. Experimental designs are sound, and data and statistical analysis supported their conclusions. Followings need to be clarified to avoid the misleading.

Page 4, Line 73, Trichomonas vaginalis should be Trichomonas vaginalis.

Corrected. Thank you.

Page 5, Line 88, Citation number (21) may be (18). Please check and correct if required.

Corrected. Thank you.

Page 6, Line 116-118,

Please clarify that parasite were inoculated first and the medium was changed after that or parasites were mixed with drug containing medium and was placed onto host cells. As they showed that invasion ratio decreases with pre-treatment of the extracellular parasites with auranofin, the time point of the initiation of drug treatment is important. The plaque number can be affected by the initial invasion, while later replication, egress, re-invasion also affects whether it becomes in visible size or not. The size of the plaque is highly recommended to be shown as an indicative value for the parasitic growth.

Yes, we agree that our plaque assay also measures any potential effect on invasion. Lines 117-119 are now edited to reflect this:

“Confluent monolayers of HFF in 6-well plates were inoculated with 1×10^2 parasites of RH *T. gondii*. Immediately after, each well received D10 containing specified concentrations of each compound.”

Page 7, Line 140/141

Please unify the terms to be used. Toxic dose (TD50) or Lethal concentration (LC50)[Table 1]

Thank you for bringing this up. We have properly edited the manuscript to show consistency. LC₅₀ in Table 1 is edited to TD₅₀.

Page 10, Line 203,

Does normalization mean that A450 values of DMSO conditions were subtracted from the treated group A450 values? Please clarify what the relative egress level represents.

The reviewer’s assumption is correct. We clarified this by adding another sentence in line 204-207:

Egress induced by each drug is presented as a relative value after subtracting control levels of LDH induced by DMSO alone (the solvent for all drugs included in this assay) from the absolute levels of LDH induced by the drugs.

Page 15, Line 318 (now line 320)

While text and table write the auranofin derivative 6 had IC₅₀ 5uM, From Fig 1B derivative 6 did not have anti-parasitic effect up to 40 uM. Please clarify.

Thank you for pointing out this discrepancy. Inadvertently, we had included preliminary data for derivative 6 only in our original manuscript. This oversight resulted in lack of error bars in Figure 1 (see below). We eliminated the sentence that derivative 6 is not effective, and lines 320-322 are now modified as follows:

“Selected auranofin derivatives 6, 10, 31, 39 with the general formula Cl–Au–PR₃ inhibited parasite growth with IC₅₀ values of 0.46 μM, 0.67 μM, 1.41 μM, 0.59 μM respectively (**Table 1**) (**Figure 1B**).”

We also updated Results section “Auranofin derivative 31 had minimal effect on the viability of host cells” (lines 325-333) to include the correct data for derivative 6:

After observing that auranofin derivatives 6, 10, 31, and 39 had similar inhibitory activity to that of auranofin against *T. gondii*, we decided to evaluate whether they have similar cytotoxicity in host cells (TD₅₀). While auranofin had an TD₅₀ of 5.14 μM, auranofin derivatives 6, 10 and 39 had TD₅₀ of 4.87 μM, 4.53 μM and 9.45 μM, respectively. Their corresponding *in vitro* therapeutic indexes, defined as the ratio of TD₅₀ on host cells over the IC₅₀ on parasites, were 10.6, 6.8, and 16, respectively. Remarkably, auranofin derivative 31 had an TD₅₀ 178 μM and an estimated *in vitro* therapeutic index of 126.2 (**Figure 1C and Table 1**).

The Discussion was modified in the following sections:

Lines 440-442: "Selected auranofin derivatives (24) 6, 10, 31, and 39 showed activity *in vitro* against *T. gondii* with similar IC₅₀ to that of auranofin (**Table 1**)."

We removed the two sentences saying the derivative 6 did not display anti-*T. gondii* activity, in lines 442 and 454.

Figure 1 Legend is modified as follows (lines 541, p.29):

Compound 6, 10, 31 and 39 inhibited *T. gondii* growth at similar IC₅₀ values. (**Table 1**).

Finally, Table has the correct IC₅₀ and TD₅₀ for derivative 6 updated (p.38).

Page 16, Line 329 (now 335): Do you mean that "Auranofin inhibited the invasion into host cells but not the replication of *T. gondii*"?

Yes, the reviewer is correct. The title is now edited as follows:

"Auranofin inhibited the invasion *into* host cells but not the replication of *T. gondii*"

Page 16, Line 336

Methods sections wrote that the invasion rate was calculated as (invaded parasites / total parasites). Or you counted the number of the host cells which had invaded parasites per total host cells? Whichever value should reflect the successful invasion event, but please clarify which is the description of what you did.

Agreed. The sentence is now rewritten as follows:

"Without pre-treatment, the percentages of *T. gondii* that successfully invaded host cells are similar in the control and auranofin treatment groups (59% vs. 60%, respectively, $p=0.7127$) (**Figure 2A**)."

We also added two more references (ref 30 and 31) in the Invasion Assay section of Methods (p.7) to show how past assays were adopted in our current study.

<https://pubmed.ncbi.nlm.nih.gov/15478810/>

<https://pubmed.ncbi.nlm.nih.gov/20435700/>

Figure 1.

Some data points seems to be missing error bars. If the experiments for the aurothioglucose, aurothiomalate, Triethyphosphine in (A), and derivative 6 in (B) was performed in single replicate, please modify the text.

We appreciate the reviewer's observation since it allowed us to detect a problem with our data for compound 6. It is now properly corrected as seen in the previous response to page 15, line 320 above.

Figure 3A

If the parasites number per vacuole are shown in raw counting numbers rather than percentage in the total pv counts, please describe how the three independent experiments are shown in this figure. Is it the representative experiments, or just the sum up of the counts from each replicates?

For each experiment, 10 random fields of view were counted. The percentage of each category with standard deviation are shown in the new figure.

Re: Spectrum02968-23R1 (**Gold(I) ion and the phosphine ligand are necessary for the anti*Toxoplasma gondii* activity of Auranofin**)

Dear Dr. Rosa M. Andrade:

You have carefully addressed all reviewers' comments.

Your manuscript has been accepted, and I am forwarding it to the ASM production staff for publication. Your paper will first be checked to make sure all elements meet the technical requirements. ASM staff will contact you if anything needs to be revised before copyediting and production can begin. Otherwise, you will be notified when your proofs are ready to be viewed.

Sincerely,
Brice Rotureau
Editor
Microbiology Spectrum